# D-Type Cyclins in Development and Disease

**DOI:** 10.3390/genes14071445

**Published:** 2023-07-14

**Authors:** Mostafa Saleban, Erica L. Harris, James A. Poulter

**Affiliations:** Division of Molecular Medicine, Leeds Institute of Medical Research, University of Leeds, Leeds LS2 9JT, UK

**Keywords:** cyclin D1, cyclin D2, cyclin D3, CDK4, cell cycle, proliferation, cancer, overgrowth

## Abstract

D-type cyclins encode G1/S cell cycle checkpoint proteins, which play a crucial role in defining cell cycle exit and progression. Precise control of cell cycle exit is vital during embryonic development, with defects in the pathways regulating intracellular D-type cyclins resulting in abnormal initiation of stem cell differentiation in a variety of different organ systems. Furthermore, stabilisation of D-type cyclins is observed in a wide range of disorders characterized by cellular over-proliferation, including cancers and overgrowth disorders. In this review, we will summarize and compare the roles played by each D-type cyclin during development and provide examples of how their intracellular dysregulation can be an underlying cause of disease.

## 1. Introduction

Progression through the cell cycle is tightly controlled by cyclin-dependent kinases (CDKs) and their regulatory partners, cyclins [1]. CDKs are activated by binding to their respective cyclin partner, followed by activating phosphorylation by a CDK-activating kinase (CAK) [2]. The transition into each phase of the cell cycle is controlled by the kinase activity of a specific cyclin-CDK complex. The ‘S-phase’ of the cell cycle initiates DNA replication, which is necessary for the forthcoming cell division [3]. Cyclin D (CCND) forms active complexes with CDK4 and CDK6 (CDK4/6), which drives the G_1_-S phase transition by phosphorylating tumour-suppressor ‘pocket proteins’ retinoblastoma (Rb), p107, and p130. Phosphorylated pocket proteins are then hyper-phosphorylated by cyclin E-CDK2 complexes, causing them to dissociate from, and release, their E2F transcription factor binding partners. This release initiates the transcription of several key proteins required in order to advance the cell cycle and promote cell proliferation [4,5]. Dysfunction of these cell cycle regulators leads to uncontrolled cell proliferation and genomic instability, contributing to tumorigenesis and overgrowth disorders [1,6,7,8].

Cyclin D-CDK4/6 can also advance the cell cycle by binding and sequestering the CDK-inhibitors (CKI) p21 and p27, which inhibits the activity of cyclin E-CDK2 complexes [9]. Outside of the Rb family, cyclin D-CDK 4/6 also phosphorylates Smad3, a transcription factor of the TGF-ß signalling pathway, and FOXM1, a transcription factor implicated in promoting cell proliferation and tumorigenesis. Phosphorylation of Smad3 at multiple sites by CDK4 inhibits its anti-proliferative response [10]. Phosphorylation of FOXM1 by CDK4/6 increases its stability, preventing cellular senescence in cancer cells as well as promoting G_1_/S cell cycle entry [11,12]. In addition to tumorigenesis, a role for Foxm1 in murine development has been described, in particular of the liver, heart, and lungs [13]. However, the Smad3 mice were viable, with death mostly occurring postnatally due to the development of neoplasms and impaired immune functions [14,15].

D-type cyclins are encoded by separate genes on three chromosomes (*CCND1*: 11q13.3, *CCND2*: 12p12.23, *CCND3*: 6p21.1) [16], and display high homology (~57% sequence identity in coding sequence) [1,17]. They can be expressed individually or in combination, typically with one D-type cyclin dominant in progressing the cell cycle of a particular cell type [17,18]. While functionally interchangeable, the loss of one or two D-type cyclins results in focused abnormalities and premature mortality in mice. Highly specialised tissues require a specific D-type cyclin [18]. Therefore, the requirement and dominance of a specific D-type cyclin in a particular tissue type is thought to be a likely result of extant transcription factors rather than the intrinsic physical properties of the cyclin [18]. Conversely, mice deficient in all three D-type cyclins die in mid-to-late gestation from heart abnormalities and severe anaemia, suggesting that cyclin D is required for the expansion of haematopoietic stem cells during development [19].

D-type cyclins serve as key endpoints of mitogenic signalling, acting as important growth factor sensors. Their transcription and activation are heavily dependent on receiving and integrating mitogenic signals from the Ras/Raf/MAPK (cyclin D synthesis), PI3K-Akt (cyclin D stability), and ß-catenin/TCF-LEF pathways [20,21,22]. Furthermore, D-type cyclins are highly labile, with proteasomal degradation occurring at rates comparable to their production. Cyclin D-CDK4/6 complexes are deactivated by the phosphorylation of a key regulatory C-terminal threonine (*CCND1*:Thr-286, *CCND2*:Thr-280, *CCND3*:Thr-283) by GSK3ß via the PI3K/AKT3/mTOR pathways, followed by export out of the nucleus, ubiquitylation in the cytoplasm by SKP1-Cullin1-F-Box (SCF) E3 ubiquitin ligase, and degradation at the proteasome [7,23,24].

Recent evidence has emerged, however, that challenges this model of cyclin D degradation. CRL4^AMBRA1^ has been identified as the E3 ubiquitin ligase that ubiquitylates cyclin D, with ubiquitylation and degradation likely occurring in the nucleus [25,26,27]. Further complicating this is the observation of cyclin D degradation independent of GSK3ß, suggesting degradation pathways which are potentially mediated by other kinases [28,29,30].

Thus, the activity of cyclin D is carefully regulated through its mitogen-activated transcription, binding to CDK4/6, phosphorylation, ubiquitylation, nuclear export, and degradation (Figure 1) [1].

## 2. Cyclin D1

Cyclin D1 (*CCND1*) encodes the general D-type cyclin, expressed in all tissue types except those derived from haematopoietic stem cell lines [16]. *CCND1* is composed of 5 exons, which are separated by 4 introns. Alternative splicing at the exon 4–intron 4 boundary results in the cyclin D1b (CCND1b) variant, which does not include exon 5, and acquires 33 new amino acids. Both isoforms are identical for the first 240 amino acids but differ in their C-termini. CCND1b is deficient in promoting Rb phosphorylation and lacks the LxxLL motif (aa 251–257) required for ligand-dependent interaction with nuclear receptors, as well as both the Thr-286 residue and the PEST sequence (aa 241–290) required for degradation (Figure 2). This is postulated to be responsible for the increased nuclear retention of the CCND1b isoform [31,32,33]. Furthermore, CCND1b has been observed to have increased oncogenic potential relative to the *CCND1* isoform. CCND1b, but not *CCND1*, was sufficient to drive the transformation of NIH3T3 cells in vitro and tumour formation in vivo [34,35]. Indeed, the CCND1b isoform is highly expressed in several cancers, including breast cancer [36], prostate cancer [37], and B-lymphoid malignancies [38].

Ccnd1-deficient mice display developmental abnormalities such as reduced body size, hypoplastic retinas, and pregnancy-insensitive mammary glands, as well as increased premature mortality within the first 3 weeks of life [23,39]. Recent studies have revealed that the mammary and retinal defects could be rescued following the knock-in of a catalytically inactive variant, Ccnd1^K112E^, indicating that Ccnd1′s role in retinal and mammary development occurs independently of CDK4/6 [40]. No human disease has yet been associated with variants in *CCND1*; however, a number of studies have identified a common *CCND1* polymorphism (c.870G>A, rs603965) that increases susceptibility to colorectal cancer and multiple myeloma [41,42,43,44].

As an oncogene, dysregulation of *CCND1* compromises the S-phase checkpoint, inducing forced progression of the cell cycle, disrupting DNA replication, and promoting DNA damage and genomic instability, resulting in oncogenesis [45]. *CCND1* is more frequently deregulated than *CCND2* and *CCND3* in both solid and haematological cancers [46], and is over-expressed and upregulated in multiple cancers, including head and neck squamous carcinoma [47], mantle cell lymphoma [48], pancreatic cancer [49], melanoma [50], non-small cell lung cancer [51], gastric cancer [52], colorectal cancer [53], endometrial cancer [54], and over 50% of human breast cancers [55]. The oncogenic activity of *CCND1* is strongly tied to its cellular levels, as tumour cells with high *CCND1* levels exhibit uncontrolled cell proliferation. Overexpression of *CCND1* can be caused by the amplification of *CCND1*, chromosomal re-arrangement, or the stabilisation of the *CCND1* protein via impaired degradation, often caused by mutations at or around the Thr-286 phosphorylation site, leading to accumulation in the nucleus [33]. Additionally, point mutations or deletions around the 3′ untranslated region (UTR) of the *CCND1* mRNA transcript result in shorter, more stable transcripts [56].

In addition to its canonical role in the Rb pathway, *CCND1* plays a key role in promoting cell proliferation, cell survival, angiogenesis, and cell migration, as well as in preventing cell senescence [17]. *CCND1* can execute some of these functions independently from its association with CDK4/6. For example, *CCND1* binds nuclear receptors such as eostrogen receptor alpha (ERα) and steroid receptor co-factors such as SRC1 and SRC3, enhancing oestrogen-receptor-mediated transcription in breast epithelial cells [11,57]. Conversely, *CCND1* inhibits the activity of androgen receptors (ARs) via binding and preventing the formation of the active AR complex, as well as recruiting histone deacetylases (HDACs) to repress its transcription [58]. Moreover, *CCND1* binds histone acetyltransferases such as p300/CREB-binding protein-associated factor (P/CAF), which increases the transcriptional activity of ER, and HDACs to enhance transcriptional repression, as seen with AR [58,59]. Indeed, *CCND1* acts as a key transcription regulator of several genes through a combination of interactions with transcription factors, coactivators, and chromatin-altering enzymes.

*CCND1* has also been shown to play a role in DNA repair. Following DNA damage, *CCND1* is recruited by BRCA2 to DNA damage sites, after which *CCND1* interacts with Rad51, a critical recombinase involved in homologous recombination and DNA repair. This facilitates its recruitment to regions of DNA damage to promote DNA repair. Decreased expression of *CCND1*, but not treatment with CDK4/6 inhibitors, decreases Rad51 recruitment to DNA damage sites, confirming that *CCND1* promotes DNA repair independently of CDK4/6 [60].

*CCND1* can also affect the movement and invasiveness of mantle cell lymphoma cells by localising and accumulating in the cytoplasm. Proteomic analysis of *CCND1*-interacting proteins showed that many of them are involved in the regulation of cytoskeletal dynamics, migration, and invasion [61]. For example, *CCND1* enhances cellular motility by inhibiting the signalling pathways of thrombospondin 1 (TSP-1) and Rho-activated kinase (ROCK) [62]. Additionally, through the inhibition of TSP1 and co-expression with the vascular growth factor VEGF, *CCND1* can promote angiogenesis, enabling tumour survival, growth, and metastasis [63]. Blocking the nuclear export of *CCND1* resulted in a significant decrease in migration and invasion, indicating that cytoplasmic *CCND1* is essential for chemotaxis and the invasion of mantle cell lymphoma cells [61].

Over-expression of *CCND1* alone, however, is not sufficient to drive oncogenic cell transformation without cooperating mutations [24]. p.Thr286Ala is a gain-of-function mutation that prevents *CCND1* from being phosphorylated by GSK3ß or AMBRA1, resulting in the stabilisation and accumulation of *CCND1* in the nucleus throughout all stages of the cell cycle. This leads to increased formation of active cyclin D-CDK4/6 complexes, genomic instability, oncogenic cell transformation, and neoplastic growth. It has been postulated that the degradation of *CCND1* is vital to maintaining genomic stability following DNA damage, and, therefore, dysregulation of its export via stabilisation is a prelude to oncogenesis rather than the underlying cause [64,65].

## 3. Cyclin D2

*CCND2*, located on chromosome 12p13, consists of 5 exons that encode the cyclin D2 (*CCND2*) protein [16]. Like *CCND1*, *CCND2* is a component of the PI3K-AKT-mTOR pathway, which controls the transition between G1 and S phase of the cell cycle by forming a complex with its cyclin-dependent kinases, CDK4 (CCND2–CDK4) and CDK6. The formation of this complex is a rate-limiting step for progression through the G1 phase, and heavily depends on the availability of serum in cells [66].

Once assembled, *CCND2*–CDK4 complexes control the phosphorylation of its substrate, Rb [67]. Phosphorylated Rb (pRb) cannot bind to the transcription factor EGF-1 [68], which increases the pool of free, active EGF protein. The accumulation of active EGF activates the expression of many downstream genes needed for S phase initiation and cell proliferation, subsequently triggering S-phase entry of the cell cycle [68,69]. A reduction in pRb, and, therefore, a reduction in the upstream production of *CCND2*, increases the amount of active (hypophosphorylated) Rb. Therefore, Rb-EGF complexes can form and prevent EGF accumulation, leading to cell cycle arrest [68]. It is, therefore, essential that the components responsible for regulating pRb production, including *CCND2*, are expressed and regulated precisely to ensure adequate entry through the cell cycle and normal cell growth.

*CCND2* is the only D-type cyclin expressed in the adult hippocampus [70], with *Ccnd2* knockout mice (*Ccnd2*-KO) displaying reduced hippocampal neuron production [71]. However, these *Ccnd2*-KO-associated brain abnormalities result in limited impact on behavioural phenotypes [72]. Although the role of *CCND2* in adult neurogenesis is less understood, there is a well-established role of *CCND2* in neurodevelopment during embryogenesis [73]. De novo germline *CCND2* mutations cause megalencephaly-polymicrogyria-polydactyly-hydrocephalus syndrome (MPPH), with mutations clustering at the c-terminus. These mutations occur at and around the phosphodegron residue Thr-280 and result in resistance to proteasomal degradation in vitro, leading to significantly increased *CCND2* accumulation in patient cells compared to controls [7]. Interestingly, the stabilisation of *CCND2* is also observed in related neurodevelopmental overgrowth conditions caused by mutations in proteins upstream of *CCND2* in the PI3K-AKT-mTOR pathway, such as AKT3, PIK3CA, and PIK3R2 [74,75,76]. *CCND2* stabilisation and/or accumulation therefore appears to be a common end point for this group of disorders, making it an excellent therapeutic target [7]. Similar results have also been observed in mouse models of brain overgrowth, with mice deficient in Dusp16, a negative regulator of MAPK, showing brain overgrowth and stabilised *CCND2* [77]. This indicates the important role that *CCND2* plays in early neurogenesis in terms of regulating cell cycle exit in neural progenitor cells, and how stabilised *CCND2* leads to the continued entering of the cell cycle into the S phase, resulting in over-proliferation and increased brain growth.

Recently, loss-of-function *CCND2* variants resulting in protein truncation were found in patients with microcephaly, the inverse brain phenotype to MPPH [78]. Prior to this study, the effect of loss-of-function mutations on *CCND2* was poorly understood, with only mice studies indicating that Ccnd2-KO causes a lack of cerebellar stellate interneurons [79], supressed adult hippocampal neurogenesis [80], and severe microcephaly [81]. These recent findings in humans with heterozygous loss of *CCND2* confirm the phenotypes observed in mice and further support a crucial role of *CCND2* during neurogenesis, as well as the requirement for the careful control of intracellular levels of cyclin D.

In addition to its key role in neurogenesis, *CCND2* also regulates the cell cycle in other tissues which may also be affected in patients with stabilising mutations. For example, MPPH patients display 3- or 4-limb postaxial polydactyly, indicating a role for *CCND2* in early limb bud development [7]. The crucial role of *CCND2* in limb development has also been identified independently in the developing limb buds of chick wings [82]. RNA sequencing of the polarising region of the limb buds and adjacent skeletal progenitor cells revealed that Ccnd2 and its inhibitor p27 are the only core cell cycle regulators expressed in these cells [83]. More recently, *CCND2* has been implicated in pancreatic B-cell proliferation, with MPPH or MCAP patients with *CCND2* or *PIK3CA* variants, respectively, found to have hypoglycaemia [84,85]. β-cells of the pancreas are responsible for the secretion of insulin; thus, over-proliferation of β-cells due to *CCND2*-stabilising mutations likely results in hyperinsulinaemia and, in turn, hypoglycaemia [86]. Patients with MPPH or MCAP are, therefore, recommended to undergo regular blood glucose monitoring, and those with low blood glucose levels should be referred for specialist endocrine review.

While its roles in development have only recently been discovered, the role of *CCND2* in cancers is more established. Prior to being identified in patients with MPPH, the same protein-stabilising mutations in and around Thr-280 had been identified somatically in tumours (COSMIC). Recent studies have identified *CCND2* and *CCND1* mutations as frequent events in myeloid leukaemia, in particular acute myeloid leukaemia [87,88]. These mutations are identical to those seen in MPPH and have the same underlying mechanisms, i.e., accumulation of stabilised *CCND2*, increased phosphorylation of Rb protein, and uncontrolled cellular proliferation.

While the accumulation of *CCND2* is the most common disease mechanism associated with *CCND2*-associated disorders, a number of cancers have been found to have reduced *CCND2* due to *CCND2* hypermethylation, particularly breast and lung cancers [89]. *CCND2*-promoter hypermethylation was found at an early stage of breast cancer tumorigenesis and was associated with the silencing of *CCND2* expression [90]. Administering the de-methylating agent antroquinonol D increased *CCND2* expression in breast cancer samples and resulted in reduced cancer cell growth through cell cycle arrest [89,91]. It remains unclear why the loss of *CCND2*, and, therefore, *CCND2*-associated cell proliferation, is observed in cancers, but this may be due to a compensation effect leading to the up-regulation of another cyclin, e.g., cyclin E. Another explanation may be related to the stage or sub-type of cancer. For example, in gastric cancer, some studies have *CCND2* hypermethylation listed as an underlying cause of proliferation [92], whereas others have found *CCND2* hypomethylation leading to increased *CCND2* expression in more advanced-stage gastric carcinomas [93].

## 4. Cyclin D3

The major isoform of *cyclin D3* (*CCND3*), located on chromosome 6p21, consists of 5 exons, but several alternative isoforms exist with different transcriptional start sites. While each isoform contains the regulatory C-terminal region of *CCND3*, the alternative start sites affect the CDK4-binding region, indicating a CDK4-independent role for these isoforms. In comparison to *CCND1* and *CCND2*, *CCND3* is predominantly found in bone marrow and lymphoid tissues, with the highest protein and RNA expression found in the thymus (proteinatlas.org) [94]. High protein expression is, however, also observed in the cerebellum, duodenum, pancreas, and testis, although RNA levels of *CCND3* in these tissues is comparatively low.

*CCND3* is an atypical D-type cyclin that is predominantly expressed in differentiated tissues [95,96,97]. Germline knockout of *CCND3* in mice is viable, but homozygous null mice show defects in lymphoid-derived cells, such as impaired B- and T-cell differentiation and granulocyte proliferation [98,99,100]. A similar role is observed in humans, with CCND3 playing a key role during B-cell precursor cell development [101] and, at later stages, a crucial role in the expansion of germinal centre B-cells [102,103].

In addition to an effect on lymphoid-derived cells, *CCND3*^−/−^ mice also show retarded growth, significant loss of muscle mass, and impaired muscle regeneration, suggesting an important role of *CCND3* in myogenesis [97,99,104]. These models suggest that, while *CCND3* may not have a direct role during development, it does play a key role in differentiation and maturation in lymphoid and musculoskeletal tissues.

Like *CCND1*, no human disorder has yet been associated with germline mutations in *CCND3*. However, a role of *CCND3* in multiple cancers has been observed, with most *CCND3* aberrations being gene-amplification rather than single-nucleotide variants. Due to the expression profile of *CCND3*, it is no surprise that lymphoid cancers are most commonly associated with dysregulation of *CCND3*. Diffuse large B-cell lymphoma (DLBCL) is the most common form of non-Hodgkin’s lymphoma, and alterations in the cyclin D/CDK4-6 pathway are found in approximately 67% of DLBCL cases, with *CCND3* overexpression accounting for 53% [105,106,107]. More significantly, DLBCL patients with high levels of *CCND3* have shown lower response rates to chemotherapy and shorter survival durations compared to those with low *CCND3* expression [108]. Increased expression of *CCND3* has also been observed in chronic lymphocytic leukaemia (CLL) cells, with RNA sequencing identifying a 38-fold increase in *CCND3* in NOTCH1-mutated cells compared to NOTCH1 non-mutated cells [109]. A similar increase was also seen for CDK4 and CDK6, and the results of a previous study associating Notch signalling with *CCND3* was confirmed [110]. A role of *CCND3* in B-cell lymphoblastic leukaemia (B-ALL) was also described, with *CCND3* being found to be indispensable for the growth and survival of B-ALL cells irrespective of the underlying driver mutation [102]. Increased *CCND3* expression associated with cells developing resistance to the CDK4 inhibitor Palbociclib, suggesting targeting of *CCND3* rather than CDK4, may be a more effective therapeutic approach. Altogether, these findings highlight a key role of *CCND3* in the regulation of lymphoid-derived cells and the consequences of its dysregulation.

In addition to lymphoid cancers, somatic mutations in *CCND3* have recently been identified in bone tumours. Xie et al. found that 43/357 (12.04%) bone tumours sequenced had a genomic aberration in *CCND3*, making it the 5th most mutated gene after *TP53*, *NCOR1*, *VEGFA,* and *RB1* [111]. *CCND3* amplifications were mainly identified in osteosarcoma patients, with 42/43 *CCND3* mutations identified being in the 227 osteosarcoma patients present in the cohort (42/227, 18.5%). A recurrent fusion gene between *KCNMB4* and *CCND3* has also been identified in a cohort of osteosarcoma, but not in a cohort of 240 other sarcomas, further suggesting a specific role of *CCND3* in osteosarcoma [112]. Functional assessment of the *KCNMB4-CCND3* fusion gene showed that it promoted cell migration in SAOS-2 cells. By comparing the ages of patients, it was found that *CCND3* mutations were more frequent in paediatric, adolescent, and young adult (P-AYA) osteosarcoma than in adult osteosarcoma [113].

## 5. Current Therapeutic Strategies for D-Type Cyclin Disorders

As D-type cyclins do not possess any enzymatic functions alone, a favoured strategy for treating cyclin D based disorders involve targeting the enzymatic activities of their partners, CDK4 and CDK6. As such, three highly specific dual-CDK4/6 inhibitors have been developed and approved by the FDA for use in the treatment of advanced or metastatic breast cancer: Palbociclib, Abemaciclib, and Ribociclib [46,114]. These inhibitors function by binding and blocking the ATP-binding pockets of CDK4 and CDK6, preventing kinase activity as well as indirect non-catalytic inhibition of CDK2 via displacement of CKI p21 [115]. More recently, these CDK4/6 inhibitors have been used to create IKZF1 and IKZF3, selective imide-based CDK4/6 degraders used to reduce cell proliferation in mantle cell lymphoma cell lines [114].

There are caveats, however: potent side effects occur due to lack of selectivity, the most notable of which is neutropenia, i.e., a low neutrophil count [11]. Furthermore, resistance to CDK4/6 inhibitors has been observed in some stabilised cyclin D phenotypes [26]. The loss of AMBRA1 can also reduce sensitivity to CDK4/6 inhibitors by stabilising cyclin D and forming active complexes with CDK2 [27]. This has led to the use of CDK4/6 inhibitors in combination with other therapies, such as hormone treatment, chemotherapy, PI3K pathway inhibitors, immunotherapy, and radiotherapy, for increased effectiveness [11,116].

Another therapeutic strategy to treat stabilised D-type cyclin disorders would be to disrupt the protein–protein interactions between cyclin D and CDK4. This would prevent the formation of active cyclin D-CDK4 complexes, arresting cell cycle progression and reducing cell proliferation in addition to also potentially inhibiting the formation of cyclin D-CDK2 complexes in some stabilised cyclin D phenotypes. However, no known inhibitors of the cyclin D-CDK4 complex have been identified. This represents an interesting opportunity to explore and investigate novel inhibitors which are able to perturb cyclin D-CDK4 complex formation.

## 6. Conclusions and Perspectives

Careful regulation of D-type cyclins is crucial to ensure there is sufficient proliferation and adequate cell numbers to develop, maintain, and repair tissues throughout life. Dysregulation of D-type cyclins, either by mutation in key regulatory proteins or through hyper-activation of upstream pathways, results in a range of disorders associated with over- or under-proliferation. Therapies that target cyclin D may offer exciting opportunities to overcome the effects of any dysregulation; however, it will be crucial to ensure that they do not tip the balance from over- to under-proliferation, or vice versa. While this review focused on the three D-type cyclins and their interactions with CDK4, it is now known that cyclin D has additional roles independent of CDK4, although what these roles are remains unclear [11,40,57,60]. Recent studies have also identified additional proteins and assembly factors required for CCND-CDK4 assembly, which will also impact downstream signalling pathways [117]. Further research is, therefore, required in order to better understand the mechanisms underpinning cyclin-D-associated regulation of the cell cycle and to identify therapeutic targets that will allow for the careful modulation of intracellular cyclin D.

## Figures and Tables

**Figure 1 genes-14-01445-f001:**
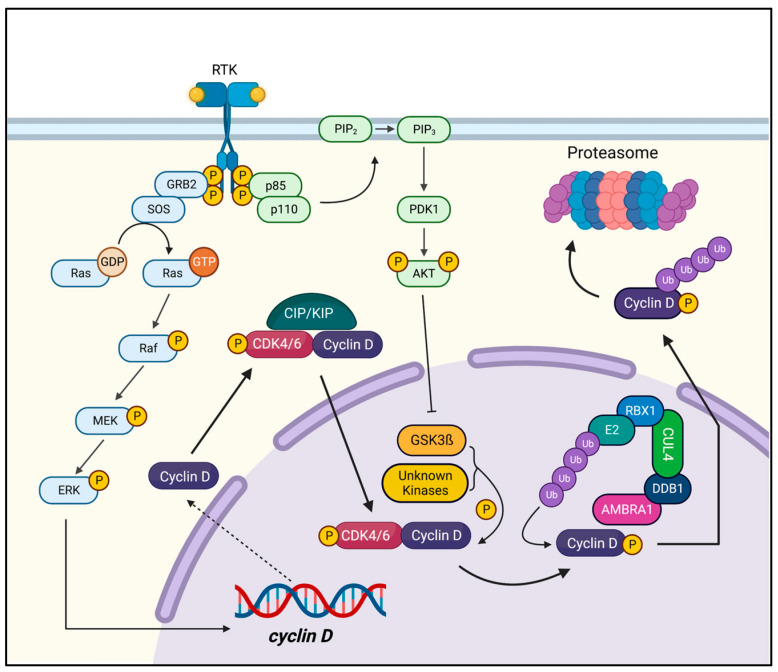
The regulation of Cyclin D: Cyclin D is expressed in response to mitogenic stimulation through the Ras (in light blue) and PI3K-Akt (in light green) pathways, which stimulate the synthesis and promote the stability of cyclin D, respectively. CDK-interacting proteins/kinase inhibitory proteins (CIP/KIP) such as p21 and p27 stabilize and facilitate the formation of the cyclin D-CDK4/6 complex. In the nucleus, the activity of this complex is terminated by the phosphorylation of C-terminal threonine by GSK3ß or, potentially, other unknown kinases. Once phosphorylated, cyclin D is then polyubiquitylated by the CRL4^AMBRA1^ E3 ubiquitin ligase and is then subsequently degraded.

**Figure 2 genes-14-01445-f002:**
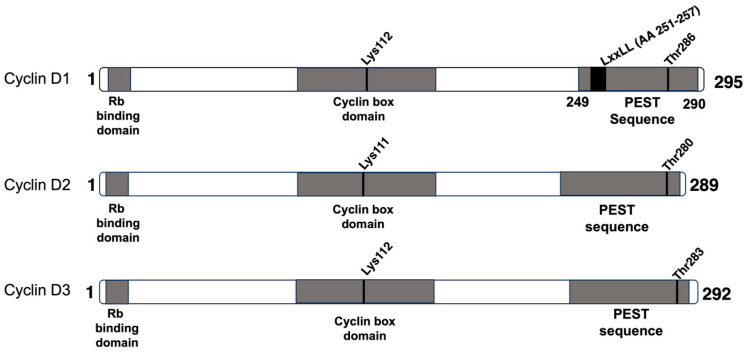
Comparison of cyclins D1, D2, and D3. The three D-type cyclins share a number of conserved sequences and domains. The Rb-binding domain, located at the N-terminus, is responsible for binding the C-terminal helix of Rb when cyclin D is bound to CDK4/6. The cyclin box domain is a heavily conserved region of ~100 amino acids located in the N-terminus of each D-cyclin, which facilitates binding to CDK4/6. Mutagenesis of Lys112 (*CCND1* and *CCND3*) and Lys111 (*CCND2*), in particular, abolishes binding to CDK4, demonstrating the residue’s essential role in mediating CDK-binding. The PEST sequence located at the C-terminus is required to mediate the degradation of cyclin D. Mutations at the Threonine phosphodegron site induce the stabilisation of cyclin D, resulting in increased progression of the cell cycle into the S-phase and genomic instability. *CCND1* also possess an LxxLL motif, which mediates ligand-dependent interaction with nuclear receptors such as Erα.

## Data Availability

Not applicable.

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
