# Peer review of "D-Type Cyclins in Development and Disease"

_genes, 2023, doi:10.3390/genes14071445_

Round 1

Reviewer 1 Report

The authors summarise and compare the roles played by each D-type cyclin during development. Also, they provide examples of how their intracellular dysregulation can be an underlying cause of disease.

This article is a review, hence it do not describe experiments.

- The information is complete and concise.

- The conclusions are consistent with the evidence and arguments presented.

- References are appropriate.

Author Response

We thank Reviewer 1 for their positive comments. No changes were made based on this review.

Reviewer 2 Report

Dear authors,

this review is an overview of D-Cyclin's role in disease and development.

The major criticism is to rewrite it in better English.

The second point in the paragraph "Outside of the Rb family, cyclin D-CDK 4/6 also phosphorylates Smad3, a transcription factor part of the TGF-ß signalling pathway and FOXM1, a transcription factor implicated in promoting cell proliferation and tumorigenesis. Phosphorylation of Smad3 at multiple sites by CDK4 inhibits its anti-proliferative response [10]. Phosphorylation of FOXM1 by CDK4/6 increases its stability, preventing cellular senescence in cancer cells, as well as promoting G1/S cell cycle entry [11, 12]", you do not mention their role in development so I ask you if these factors have only a role in cell proliferation and tumorigenesis?

The third point regards 38X is better to write a 38-fold increase.

The fourth point is about the osteosarcoma cohort can you insert the number of osteosarcomas?

The last point is regarding the conclusion that is clear but may be improved, perspectives in particular.

Dear authors,

this review is an overview of D-Cyclin's role in disease and development.

The major criticism is to rewrite it in better English.

Author Response

We thank reviewer 2 for their comments and suggestions. A point-by-point response can be found below:

1) The major criticism is to rewrite it in better English.

We have taken on board the reviewers’ comments of the written English and have improved this throughout.

2) The second point in the paragraph "Outside of the Rb family, cyclin D-CDK 4/6 also phosphorylates Smad3, a transcription factor part of the TGF-ß signalling pathway and FOXM1, a transcription factor implicated in promoting cell proliferation and tumorigenesis. Phosphorylation of Smad3 at multiple sites by CDK4 inhibits its anti-proliferative response [10]. Phosphorylation of FOXM1 by CDK4/6 increases its stability, preventing cellular senescence in cancer cells, as well as promoting G1/S cell cycle entry [11, 12]", you do not mention their role in development so I ask you if these factors have only a role in cell proliferation and tumorigenesis?

To clarify this point from the reviewer, we have added the following text at the end of the paragraph (line 39 onwards): “In addition to tumorigenesis, a role for Foxm1 in murine development has been described, in particular of the liver, heart and lungs [13], however Smad3 mice are viable with death mostly occurring postnatally due to the development of neoplasms and impaired immune functions [14, 15].”

3) The third point regards 38X is better to write a 38-fold increase.

We have updated the text on line 286 to “identifying a 38-fold increase”

4) The fourth point is about the osteosarcoma cohort can you insert the number of osteosarcomas?

We have updated the text from line 298 to read “CCND3 amplifications were mainly identified in osteosarcoma patients, with 42/43 CCND3 mutations identified being in the 227 osteosarcoma patients present in the cohort (42/227, 18.5%).”

5) The last point is regarding the conclusion that is clear but may be improved, perspectives in particular.

We have updated the conclusions and perspectives paragraph to include more perspectives in the field from line 349.

Round 2

Reviewer 2 Report

Dear authors,

I have carefully read your review and it is a pleasure to tell you that the work is fine and ready to be published.

Best regards